# The Effect of Donors’ Demographic Characteristics in Renal Function Post-Living Kidney Donation. Analysis of a UK Single Centre Cohort

**DOI:** 10.3390/jcm8060883

**Published:** 2019-06-20

**Authors:** Maria Irene Bellini, Sotiris Charalampidis, Ioannis Stratigos, Frank J.M.F. Dor, Vassilios Papalois

**Affiliations:** 1Renal and Transplant Directorate, Imperial College Healthcare NHS Trust, W120HS London, UK; s.charalampidis@nhs.net (S.C.); Frank.dor@nhs.net (F.J.M.F.D.); vassilios.papalois@nhs.net (V.P.); 2Queen Mary University of London, E14NS London, UK; i.stratigos@se16.qmul.ac.uk; 3Department of Surgery and Cancer, Imperial College London, SW72AZ London, UK

**Keywords:** living donor, kidney transplantation, ethnicity, age, obesity, genetic relationship donor/recipient

## Abstract

Introduction: There is a great need to increase the organ donor pool, particularly for living donors. This study analyses the difference in post-living donation kidney function according to pre-donation characteristics of age, genetic relationship with the recipient, sex, ethnicity, and Body Mass Index (BMI). Methods: Retrospective single centre analysis of the trajectory of estimated Glomerular Filtration Rate (eGFR) post-living kidney donation, as a measure of kidney function. Mean eGFR of the different groups was compared at 6 months and during the 60 months follow up. Results: Mean age was 46 ± 13 years, 57% were female, and 60% Caucasian. Mean BMI was 27 ± 5 kg/m^2^, with more than a quarter of the cohort having a BMI > 30 (26%), and the majority of the donors genetically related to their recipients (56%). The higher decline rate in eGFR was at 6 months after donation, with female sex, non-Caucasian ethnicity, and age lower than 60 years being independently associated with higher recovery in kidney function (*p* < 0.05). In the 60 months follow up, older age, genetic relationship with the recipient, and male sex led to higher percentual difference in eGFR post-donation. Conclusion: In this study, with a high proportion of high BMI living kidney donors, female sex, age lower than 60 years, and non-genetic relationship with recipient were persistently associated with higher increase in post-donation kidney function. Ethnicity and BMI, per se, should not be a barrier to increasing the living donor kidney pool.

## 1. Introduction

Living donor (LD) kidney transplantation provides the best long-term outcomes for patients with chronic kidney failure [1]. A careful selection to limit the potential risks related to living kidney donation is important, not only to safeguard these healthy individuals who should not be harmed as a result of their generous act, but also to keep expanding the organ donor pool [2]. The donor’s demographic characteristics are currently a topic of interest to assess the potential risk of end stage renal disease (ESRD) among living donors. 

It has been reported that high body mass index (BMI kg/m^2^) has no higher perioperative risks for living kidney donors [3] but has a negative impact on post-donation kidney function [4]. However, there is no consensus regarding the BMI threshold for LD acceptance criteria; this is particularly important in populations where the average BMI is increasing [5,6]. 

Previous studies have also suggested that renal function reached at one year post-donation remains stable—at least over the next decade—but then declines with ageing [7,8]. Ibrahim et al. indicated that a younger age at the time of donation, a longer time since donation, and a higher eGFR at the time of donation were associated with a greater compensatory increase in the eGFR in the remaining kidney [9]. Conversely, Dols et al. reported that kidney donation by older donors is relatively safe over time since, in their experience, kidney function did not decline progressively [10,11]. A correlation with age and ethnicity has also been reported, with higher risk for ESRD for older Caucasians and younger Africans [12].

Regarding the donor’s sex, there are reports that in males there is a more pronounced decline in the short-term renal function [13], but the absolute risk to develop ESRD is still very low: one per 2000, compared to one per 10000 in the general population (RR 8.83), according to a recent meta-analysis [14]. 

Living kidney donors with a first-degree genetic relationship to the recipient have an increased risk of developing ESRD [15], but a personalised estimation on the basis of donor characteristics still remains unavailable and controversial.

Risk estimation is critical for appropriate informed consent, so special consideration in potential living donors’ relative risk triggered by certain demographics or conditions is massively important nowadays. This single centre study aims to establish the effect of the donor’s sex, age, ethnicity, BMI, and genetic relationship to the recipient on the evolution of the eGFR as a marker of kidney function recovery after living kidney donation.

## 2. Methods

The study, performed in accordance with the Declaration of Helsinki principles, is a retrospective analysis on consecutive living kidney donors who had their operation in our centre during the period of 2000 and 2017, and with 60 months of follow up. After discharge, donors first came to our follow up clinic in 2 weeks’ time, and then at six months and yearly (up to five years) post-donation for routine blood controls.

The data used were anonymised and extracted from an electronic database of medical records. The study fell under the category of research through the use of anonymised data of existing databases which, based on the Health Research Authority criteria [16], does not require proportional or full ethics review and approval.

Obesity was defined according to the World Health Organization (WHO) classification when BMI ≥ 30 kg/m^2^, normal weight when BMI ≤ 25 kg/m^2^, and overweight when 25 kg/m^2^ < BMI < 30 kg/m^2^. Donors were also stratified according to sex, ethnicity, and age below or above 60 years, which is the cut off between standard or extended criteria used in deceased donor organ donation [17]. Mean eGFR was compared using the CKD-EPI equations between groups, as this is recommended in view of the eligibility of potential living donors [18]. The decline in kidney function was analysed between different points of the 60 months follow up and donation. It was expressed as the percentual difference in eGFR (Δ eGFR), between mean eGFR at a given point of follow up and the eGFR at the time of donation.

Continuous variables were presented as mean ± standard deviation and compared using one-way ANOVA at 6, 12, 24, 36, 48, and 60 months follow up. Confidence interval was set to 95%, and *p* was considered significant at less than 0.05. A general linear model of repeated measures of ANOVA eGFR during the 60 months follow up was built to observe the percentual difference in eGFR recovery and assess the independent effect of donors’ characteristics. 

Statistical analysis was performed using SPSS (IBM SPSS Statistics for Windows, Version 20.0; IBM Corp, Armonk, NY, USA). 

## 3. Results

A total of 889 consecutive living kidney donors were analysed (Table 1). Mean follow up was 44.1 ± 31.3 months. Full dataset at 6 months was available for 700 donors (79%), at 12 months for 635 (71%), at 24 months for 569 (64%), at 36 months for 489 (55%), at 48 months for 408 (46%), and at 60 months for 348 (39%) donors. The mean age at donation was 46 ± 13 years, with a mean BMI of 27 ± 5 kg/m^2^. More than a quarter of the total LD cohort had a BMI > 30 (26%). Females were 57% and the prevalent ethnicity was Caucasian (60%). The majority of the donors were genetically related to the recipient (56%).

Mean eGFR at donation was confirmed to be statistically significantly related to sex, age, and ethnicity (*p* < 0.001), but not to the BMI or the genetic relationship with the recipient (Table 1). 

More specifically, females, donors aged > 60 years, and Caucasians had lower eGFR at donation. Mean eGFR was compared in the different groups during follow up, evaluating the evolution of kidney function recovery after donation. It was expressed as Δ eGFR, between mean eGFR at a given point of follow up and the eGFR at the time of donation. The lowest mean eGFR was after 6 months from donation (*p* < 0.001), with then a progressive decrease of Δ eGFR in the 60 months of follow up, mirroring a progressive recovery in kidney function after donation. Δ eGFR for the whole cohort at 6 months was 0.27 ± 16, Δ eGFR at 12 months 0.26 ± 15, Δ eGFR at 24 months 0.24 ± 15, Δ eGFR at 36 months 0.22 ± 16, Δ eGFR at 48 months 0.2 ± 17, and Δ eGFR at 60 months 0.18 ± 17 (*p* < 0.001). See Appendix A for difference in Δ eGFR within different groups.

At 6 months of follow up, the nadir of the kidney function recovery, age older than 60 years, male sex, and Caucasian ethnicity were independent predictors for lower recovery in kidney function (*p* < 0.05). No effect was observed for different classes of BMI and genetic relationship with the recipient. 

Figure 1, Figure 2, Figure 3, Figure 4 and Figure 5 represent mean eGFR and the percentual difference from the donation eGFR during the 60 months follow up. The generalised repeated measures of ANOVA eGFR and Δ eGFR are stratified according to donor age, relationship with the recipient, sex, BMI, and ethnicity. 

Mean eGFR post-donation was confirmed as persistently higher in the younger cohort (*p* < 0.001, Figure 1A). The percentual difference in eGFR during the 60 months follow up was also significantly different, with lower recovery from the original kidney function for donors aged > 60 years (*p* = 0.037, Figure 1B).

Figure 2A shows that although the genetic relationship with the recipient did not statistically affect mean eGFR during the follow up (*p* = 0.168), there was a difference in the kidney function recovery after donation (Figure 2B), in favor of living unrelated donors (*p* = 0.007).

Males had higher mean eGFR at donation (*p* > 0.001), but there was no significant difference in mean eGFR after donation (*p* = 0.3, Figure 3A). The recovery of kidney function was persistently higher in females (*p* < 0.001, Figure 3B). This aspect could also be noted in Appendix A, where only at donation time male donors have a higher mean eGFR: 95 ± 32 mL/min/1.72 m^2^ versus 87 ± 22 mL/min/1.72 m^2^ for female donors (*p* < 0.001), and where the Δ eGFR is persistently higher for men during follow up (*p* < 0.001).

Figure 4A,B represent kidney function according to different BMI classes; there was no significant difference pre- or post-donation, either in mean eGFR (*p* = 0.53) or in the percentual difference in kidney function recovery (*p* = 0.79) observed during the study period.

Finally, Caucasian ethnicity was related to a lower mean eGFR pre- and post-donation (*p* < 0.001, Table 1 and Figure 5A). There was also an ethnicity-related effect in the early kidney function recovery after donation at 6 months (*p* = 0.005) in favor of Africans and Asians, see also Appendix A. However, this ethnicity-related effect in kidney function recovery was not observed during the 60 months follow up, with no significant different mean Δ eGFR in the general linear model (*p* = 0.38, Figure 5B).

In a logistic regression with stepwise procedure, it was confirmed that only male sex and older age affected Δ eGFR in the long term.

## 4. Discussion

The present study investigated the kidney function recovery post-living donation in our centre at 6 months post-donation, and during 60 months follow up. Donors’ demographic characteristics of age, genetic relationship to the recipient, sex, BMI, and ethnicity were analysed, aiming to assess long term risk, offer comprehensive information to potential donors, and further encourage living kidney donation [19]. 

Sixty percent of the living kidney donors in our study were Caucasian, the absolute majority for the cohort, but an inferior percentage compared to the rest of the UK, where 88% of the entire living donor pool are Caucasian [19]. The lack of an ethnicity-related effect to the recovery of kidney function could encourage potential living donors from African and Asian communities to proceed with donation; this is most important, considering the fact that these minority groups are more likely to deny consent for deceased organ donation, while at the same time they face prolonged waiting times due to difficulties in the matching process independently from the allocation policy [18,19]. In our centre, practicing in a multi-ethnic country like the UK, we focus on educational programmes directed to ethnic minorities to assure potential donors of the long-term safety of living kidney donation; the results of this study and similar findings in other cohorts [20,21] are informing the content of those programmes.

Similarly to other studies, we demonstrated that the lowest eGFR is within 6 months of follow up [20], and it is statistically significantly related to age older than 60 years [21] and male sex [15]. It is interesting to note that the mean eGFR is higher pre-donation for males, and after donation there is no difference during the 60 months follow up. This happens in relation to the higher Δ eGFR for the males versus females expressing their living donor kidney function recovery. The general longer life expectancy for women could also be a contributing factor [22].

Previous studies have demonstrated the overall safety of living kidney donation, even at older ages [10]. In our study, 15% of LDs were > 60 years old. Our data showed that the recovery of kidney function following living kidney donation was lower for the elderly, reflecting most probably the natural biological process [23]. However, the final mean eGFR for donors > 60 years old was 54 ± 11 mL/min/1.73 m^2^, a very satisfactory outcome after 60 months of follow up (Figure 1A,B). Therefore, our centre policy is that living donation is not discouraged on the basis of age only.

Whether or not obesity plays a role in the deterioration of the kidney function following living kidney donation remains a hotly debated issue. Despite the recent report from Locke et al. of a 1.9-fold higher risk for ESRD when compared to normal BMI donors [4], the adjusted risk of ESRD associated with obesity is only 1.16 in living donors with obesity [24]. The 26% of living donors in our cohort had a BMI > 30, but there was no increased risk in terms of mean eGFRs and Δ eGFR or recovery in kidney function during the 60 months follow up (Figure 4A,B). As it is the case with age, our centre policy does not rely for LD risk on the basis of BMI only: Potential candidates are screened on the basis of a multi-disciplinary input and discussion with a multifactorial analysis, tailored on a case-by-case basis to consider the overall medical condition, rather than a single factor [25]. In our centre, we screen potential donors on the basis of a multi-disciplinary input and discussion. We investigate the past medical history, previous hospitalisation, and regular medications. Hypertension controlled with up to two medications is not a contraindication according to the British Transplant Society guidelines [26]; moreover, there is evidence that this does not affect long-term overall or cardiovascular mortality [27]. Proteinuria, diabetes, angina/heart disease, stroke, severe pulmonary disease, kidney stone (or other kidney disease), blood disorders, sickle cell disease, or active cancer are contraindications to donate. No BMI or age cut off are considered as standing only criteria to proceed with donation. All the potential donors undergo imaging of their kidneys and then decision to proceed is made if eGFR > 80 mL/min/1.73 m^2^.

Finally, we found a statistically significant higher Δ eGFR after donation for genetically related donors (LRDs), as shown in Figure 2B. The concern that biologically-related living donors, especially first-degree relatives, may face increased risk of adverse renal outcomes after living kidney donation has been a topic of discussion in the transplant community for a long time [28,29]. Increased risks of renal failure in close biological relatives of ESRD patients have been observed in population-based and case-control studies among non-donors, regardless of whether recipient ESRD has a known hereditary cause [30,31]. In a study of Caucasian donors in Norway, the nine donors who developed ESRD were all biologically related to their recipients, and renal failure appeared mainly due to immunological diseases [21]. We believe that, although there is still controversy in the literature around whether LRDs are at increased risk compared to Living Unrelated Donors (LURDs) [32,33], the difference in mean eGFR after donation demonstrated in our study generates the responsibility for the transplant team to inform potential donors genetically related to the recipient about the potential increased risk of ESRD. A possible explanation for this finding, along with the latency of some genetic renal disorders running in the family, could be the presence of emotional factors [34] that push towards the decision to donate in a biologically related donor more than in an unrelated donor. It is also possible that the background risk of ESRD is dependent on several conditions, including family habits or life in a particular socio-economic area—and together, all of these may affect synergistically the eGFR trajectory more significantly than in unrelated donors.

The retrospective nature of this study has limited our analysis to those patients for whom we had complete data during the follow up period. The difficulty for a lifelong follow up in living kidney donors is possibly related to the fact that they are healthy individuals, therefore reluctant to come for tests after the first year post-donation. Another possible explanation could be the fact that as the years go by, donors move around the country and there is no continuity in their follow up.

## 5. Conclusions

The present study demonstrated that the higher decline rate in eGFR post-living kidney donation was at 6 months, with female sex, non-Caucasian ethnicity, and age lower than 60 years being independently associated with higher recovery in kidney function. At 60 months follow up, older age, genetic relationship with the recipient and male sex were related to a lesser recovery of eGFR, with male sex and older age confirmed at logistic regression analysis. BMI did not relate to kidney function pre or post living kidney donation.

## Figures and Tables

**Figure 1 jcm-08-00883-f001:**
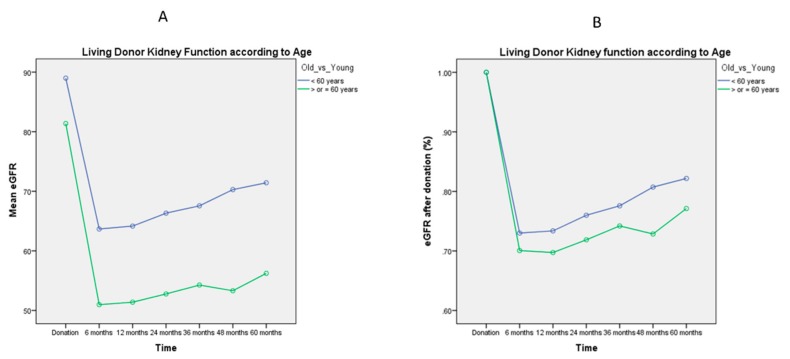
General Linear Model of Repeated Measures ANOVA of mean eGFR (Figure 1A) and mean Δ eGFR (Figure 1B) during the 60 months follow up. There is a decline in eGFR post-donation, with a statistical significant correlation with age > 60 years (*p* < 0.001). The percentual difference in eGFR is also statistically different, with lower recovery for age > 60 years (*p* = 0.037).

**Figure 2 jcm-08-00883-f002:**
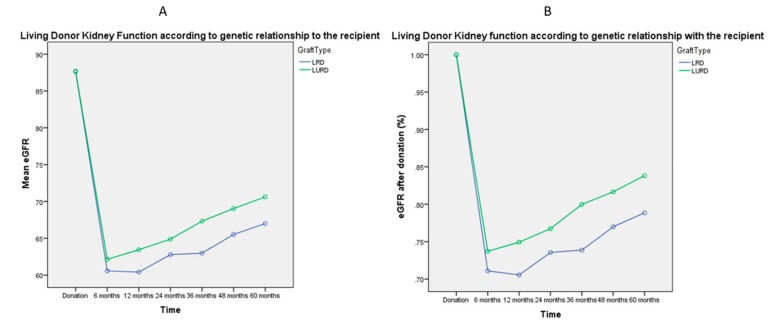
General Linear Model of Repeated Measures ANOVA of mean eGFR (Figure 2A) and mean Δ eGFR (Figure 2B) during the 60 months follow up. There is no difference in mean eGFR post-donation according to the genetic relationship with the recipient (*p* = 0.168), while the percentual recovery in eGFR is statistically different, being higher for live unrelated donor (*p* = 0.007). LRD: Living Related Donor; LURD: Living Unrelated Donor.

**Figure 3 jcm-08-00883-f003:**
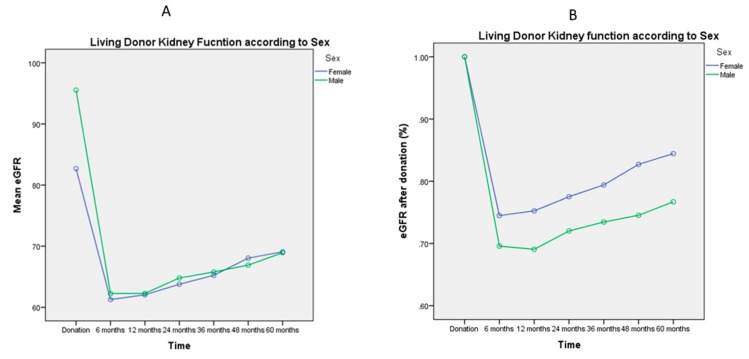
General Linear Model of Repeated Measures ANOVA of mean eGFR (Figure 3A) and mean Δ eGFR (Figure 3B) during the 60 months follow up. Mean eGFR at donation is lower in females (*p* < 0.001), but in the 60 months follow up mean eGFR is no longer significant (*p* = 0.3), because the mean percentual difference in eGFR is higher in males (*p* < 0.001).

**Figure 4 jcm-08-00883-f004:**
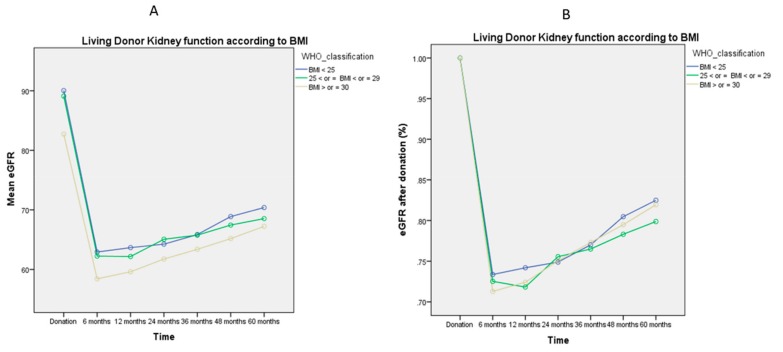
General Linear Model of Repeated Measures ANOVA of mean eGFR (Figure 4A) and mean Δ eGFR (Figure 4B) during the 60 months follow up. The mean eGFR post-donation does not differ according to BMI (*p* = 0.53), with no difference also in kidney function recovery after live donation (*p* = 0.79).

**Figure 5 jcm-08-00883-f005:**
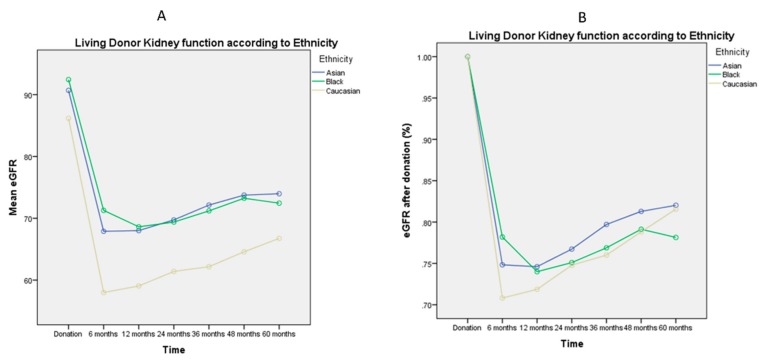
General Linear Model of Repeated Measures ANOVA of mean eGFR (Figure 5A) and mean Δ eGFR (Figure 5B) during the 60 months follow up. The mean eGFR post-donation confirmed to be lower for Caucasian ethnicity (*p* < 0.001), as well as the recovery in kidney function at 6 months (*p* = 0.035). There is not a statistically significance difference in the general linear model for the Δ eGFR at 60 months follow up (*p* = 0.38).

**Table 1 jcm-08-00883-t001:** Baseline demographic characteristics and eGFR reported in mL/min/1.73 m^2^ at donation. In bold are highlighted statistically significant higher values. LRD: Living Related Donor; LURD: Living Unrelated Donor.

	eGFR Donation		
	Mean ± St. Dev.	*p* Value	Table *N* %
Sex			
Female	**87 ± 22**	**<0.001**	57
Male	**95 ± 32**		43
Ethnicity			
Asian	**91 ± 19**	**<0.001**	25
African	**103 ± 23**		15
Caucasian	**87 ± 28**		60
Graft Type			
LRD	91 ± 26	0.94	56
LURD	90 ± 29		44
Age (years)	46 ± 13		100
< 60 years	**93 ± 24**	**<0.001**	85
> or = 60 years	**80 ± 39**		15
WHO classification BMI (kg/m^2^)	27 ± 5		100
BMI < 25	92 ± 29	0.5	36
25 < or = BMI < or = 29	91 ± 30		38
BMI > or = 30	89 ± 20		26

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
