# Peer review of "The Effect of Donors’ Demographic Characteristics in Renal Function Post-Living Kidney Donation. Analysis of a UK Single Centre Cohort"

_jcm, 2019, doi:10.3390/jcm8060883_

Reviewer 1 Report

It is not clear in Materials and methods, how GFR is calculated. Authors should state in the discussion, which method they preferred and why, considering also briefly pro and cons

Author Response

Dear Editor,

Dear Reviewer,

We are most grateful for your review of the manuscript and your helpful comments.

Please kindly see our replies to your comments which have also been incorporated in the manuscript.

We are at your disposal should you have any further questions.

Yours truly,

Maria Irene Bellini MD, PhD, FEBS

Reviewer 1:

It is not clear in Materials and methods, how GFR is calculated. Authors should state in the discussion, which method they preferred and why, considering also briefly pro and cons

Thank you.  We used CKD-EPI formula as this is the one recommended for eligibility of potential living donors. We added this in the methods and specified the reference: Gaillard F, Courbebaisse M, Kamar N, Rostaing L, Jacquemont L, Hourmant M, Del Bello A, Couzi L, Merville P, Malvezzi P, Janbon B, Moulin B, Maillard N, Dubourg L, Lemoine S, Garrouste C, Pottel H, Legendre C, Delanaye P, Mariat C. Impact of estimation versus direct measurement of pre-donation glomerular filtration rate on the eligibility of potential living kidney donors.  Kidney Int. 2019 Apr; 95(4):896-904. doi: 10.1016/j.kint.2018.11.029. Epub 2019 Feb 26.

Reviewer 2 Report

A well-presented paper and an important area of research. 

I think the use of eGFR in this varied population can be problematic with significant error range regardless of the formula used. I think this needs acknowledgment and discussion. Does the data change if a weight based eGFR measurement is used especially for BMI category or just serum creatinine?  Are the differences noted clinically significant especially with respect to the in eGFR or just a reflection of the “correction” factors used for male v female, black v Caucasian and age. I’m sure a more accurate measure of renal function was performed prior to donation and this could be incorporated in the baseline data and compared to the eGFR at baseline as a “quality control”.

With follow up data there could be meaningful data included on proteinuria and hypertension. I think there needs to be some information about the use of antihypertensives especially ACE or ARB as this can of course impact on eGFR.

Finally, the comment about the differences noted between genetic relationship to donor needs expansion. It seems counterintuitive and the explanation needs expansion.

Author Response

Dear Editor,

Dear Reviewer,

We are most grateful for your review of the manuscript and your helpful comments.

Please kindly see our replies to your comments which have also been incorporated in the manuscript.

We are at your disposal should you have any further questions.

Yours truly,

Maria Irene Bellini MD, PhD, FEBS

A well-presented paper and an important area of research. 

I think the use of eGFR in this varied population can be problematic with significant error range regardless of the formula used. I think this needs acknowledgment and discussion. Does the data change if a weight based eGFR measurement is used especially for BMI category or just serum creatinine?  Are the differences noted clinically significant especially with respect to the ∆ in eGFR or just a reflection of the “correction” factors used for male v female, black v Caucasian and age. I’m sure a more accurate measure of renal function was performed prior to donation and this could be incorporated in the baseline data and compared to the eGFR at baseline as a “quality control”.

Thank you. We actually look predominantly at the eGFR when considering potential living donor candidates. We also agree with the recent paper  by Gaillard F, Courbebaisse M, Kamar N, Rostaing L, Jacquemont L, Hourmant M, Del Bello A, Couzi L, Merville P, Malvezzi P, Janbon B, Moulin B, Maillard N, Dubourg L, Lemoine S, Garrouste C, Pottel H, Legendre C, Delanaye P, Mariat C. Impact of estimation versus direct measurement of predonation glomerular filtration rate on the eligibility of potential living kidney donors. Kidney Int. 2019 Apr;95(4):896-904. doi: 10.1016/j.kint.2018.11.029. Epub 2019 Feb 26.

With follow up data there could be meaningful data included on proteinuria and hypertension. I think there needs to be some information about the use of antihypertensives especially ACE or ARB as this can of course impact on eGFR.

Thank you, we did not focus on hypertension as it does not impact the long term overall mortality or cardiovascular risk for living donors: Haugen AJ, Langberg NE, Dahle DO, Pihlstrøm H, Birkeland KI, Reisaeter A, Midtvedt K, Hartmann A, Holdaas H, Mjøen G. Long-term risk for kidney donors with hypertension at donation - a retrospective cohort study. Transpl Int. 2019 Apr 15. doi: 10.1111/tri.13443.

Proteinuria is a contraindication to donation, therefore if detected at the moment of the donor candidates screening, they are not considered suitable at our centre.

We modified as following: Hypertension controlled with up to two medications is not a contraindication to donation according to the British transplant Society guidelines, moreover there is evidence that this does not affect long-term overall or cardiovascular mortality; proteinuria, diabetes, angina/heart disease, stroke, severe pulmonary disease, kidney stone (or other kidney disease), blood disorders, sickle cell disease or active cancer are contraindications to donate.

Finally, the comment about the differences noted between genetic relationship to donor needs expansion. It seems counterintuitive and the explanation needs expansion.

Thank you, we modified as following: A possible explanation for this finding, along with the latency of some genetic renal disorders running into the family, could be the presence of emotional factors that push towards the decision to donate in a biologically related donor more than in an unrelated. It is also possible that the background risk of ESRD is dependent from several conditions, including family habits or life in a particular socio-economic area and these all together may affect synergistically the eGFR trajectory more significantly than in unrelated donors.

Reviewer 3 Report

Overall comments to the Authors

Thank you for the opportunity to review the manuscript entitled, "The effect of donor’s demographic characteristics in renal function post living kidney donation. Analysis of a UK single centre cohort.". I think that the effect of donor’s demographic characteristics on the post-donation renal function worth investigating. This report is interesting and useful in this field. And also, the manuscript is well written, in total.

I have the following comment:

Major

1. As the authors stated, more than the half of the data were unavailable during long-term follow up, which attenuates the significance of the conclusions.

2. The authors concluded that older age, LRD, and male sex were associated with impaired recovery of renal function based on the univariate analysis. Multivariate analysis should be performed in order to assess the independent contributing factors.

Minor

1.     In the whole manuscript, the authors need to correct literal errors, e.g. incorrect size of the word in line 55, missingness of space before the quotation in line 68 and 201, and Figure 2A, 2B, not 2a, 2b in line 127.

2.     In the Discussion, the clarification of p value is unnecessary, line 209.

Author Response

Dear Editor,

Dear Reviewer,

We are most grateful for your review of the manuscript and your helpful comments.

Please kindly see our replies to your comments which have also been incorporated in the manuscript.

We are at your disposal should you have any further questions.

Yours truly,

Maria Irene Bellini MD, PhD, FEBS

Thank you for the opportunity to review the manuscript entitled, "The effect of donor’s demographic characteristics in renal function post living kidney donation. Analysis of a UK single centre cohort.". I think that the effect of donor’s demographic characteristics on the post-donation renal function worth investigating. This report is interesting and useful in this field. And also, the manuscript is well written, in total.

I have the following comment:

Major

1.     As the authors stated, more than the half of the data were unavailable during long-term follow up, which attenuates the significance of the conclusions.

Thank you. Life-long follow up for living donors is a challenge due to the fact that, as the years go by, these healthy individuals do not feel the necessity to come to the hospital for check-up. We do acknowledge this limitation, although given that the highest decline in eGFR is within the first year, we still have consistent data to support our analysis.

The authors concluded that older age, LRD, and male sex were associated with impaired recovery of renal function based on the univariate analysis. Multivariate analysis should be performed in order to assess the independent contributing factors.

Thank you, we run a logistic regression with stepwise procedure and only gender and age affect ΔeGFR, confirming the findings reported. We added this in the manuscript as well.

Minor

1.        In the whole manuscript, the authors need to correct literal errors, e.g. incorrect size of the word in line 55, missingness of space before the quotation in line 68 and 201, and Figure 2A, 2B, not 2a, 2b in line 127.

Thank you, we have corrected as per suggestion.

2.     In the Discussion, the clarification of p value is unnecessary, line 209.

Thank you, we have removed it.

Round  2

Reviewer 2 Report

Happy with proposed changes. 

Author Response

Thank you

Reviewer 3 Report

The authors correctly answered my questions. As a result of multivariate analysis, the conclusions might be able to be revised. Otherwise, I have no more comments.        

Author Response

Thank you, we changed the conclusions as following: The present study demonstrated that the higher decline rate in eGFR post living kidney donation was at 6 months, with female sex, non-Caucasian ethnicity and age lower than 60 years, being independently associated with higher recovery in kidney function. At 60 months follow up, older age, genetic relationship with the recipient and male sex were related to a lesser recovery of eGFR, with male sex and older age confirmed at logistic regression analysis. BMI did not relate to kidney function pre or post living kidney donation.